# Analysis of Pulp Tissue Viability and Cytotoxicity of Pulp Capping Agents

**DOI:** 10.3390/jcm12020539

**Published:** 2023-01-09

**Authors:** Pratima Panda, Shashirekha Govind, Sanjit Kumar Sahoo, Satabdi Pattanaik, Rachappa M. Mallikarjuna, Triveni Nalawade, Sanjay Saraf, Naseer Al Khaldi, Salma Al Jahdhami, Vinay Shivagange, Amit Jena

**Affiliations:** 1Department of Conservative Dentistry and Endodontics, Institute of Dental Sciences, Siksha ‘O’ Anusandhan (Deemed to be) University, Bhubaneswar 751003, India; 2Department of Paediatric Dentistry, Child Dental Health, Oman Dental College, Mina Al Fahal, Muscat 116, Oman; 3Department of Oral Biology and Oral Pathology, Oman Dental College, Mina Al Fahal, Muscat 116, Oman; 4Consultant Prosthodontist, Al-Nahdha Hospital, Al Khuwair, Muscat 133, Oman; 5Oral and Maxillofacial Surgeon, Al-Nahdha Hospital, Al Khuwair, Muscat 133, Oman; 6Department of Conservative Dentistry and Endodontics, Saveetha Dental College, Chennai 600077, India; 7Department of Conservative Dentistry and Endodontics, Sriram Chandra Bhanja Dental College & Hospital, Cuttack 753007, India

**Keywords:** vital pulp therapy, human dental pulp stem cells, MTA, Theracal PT, Tetric N-Bond Bonding agent, PRF

## Abstract

The present research study assessed the cell viability and cytotoxic effect of mineral tri-oxide aggregate (MTA), Tetric N-Bond Universal bonding agent, Theracal PT (pulpotomy treatment), and platelet-rich fibrin (PRF) as pulp capping agents on human dental pulp stem cells (hDPSCs). The cells were isolated from the pulp tissue of an extracted healthy permanent third molar. After four passages in Dulbecco’s Modified Eagle’s Medium, the primary cells were employed for the investigation. The test materials and untreated cells (negative control) were subjected to an Methylthiazol-diphenyl-tetrazolium (MTT) cytotoxicity assay and assessed at 24-, 48-, and 72-h intervals. The Wilcoxon matched-paired *t*-test and Kruskal–Wallis analysis of variance (ANOVA) test were applied (*p* < 0.05). PRF imparted the highest cell viability at 48 h (*p* < 0.001), followed by MTA, Theracal PT, and Tetric N-Bond. Similarly, PRF had the highest potential to enhance cell proliferation and differentiation (*p* < 0.001), followed by Theracal PT, MTA, and the bonding agent at the end of 24 h and 72 h, respectively. Finally, PRF sustained the viability of human primary dental pulp stem cells more effectively than Theracal PT and MTA; however, the application of a Tetric N-Bond as a pulp capping agent was ineffective.

## 1. Introduction

Direct pulp capping (DCP) is regularly performed following deep caries treatment to stimulate reparative dentin, a physiological barrier that acts as a “biological seal” to protect and preserve the underneath pulp tissue’s vitality [1]. Direct pulp capping materials are biocompatible and have antibacterial properties. The most important characteristics of these materials include the promotion of tissue healing, cytocompatibility, and the ability to seal the lesion [2,3,4]. The biomaterial employed in this therapy is generally used to encourage the production of mineralized tissue by using pulp cells [5]. The goal of a regenerative treatment for direct pulp capping is to stimulate odontoblast-like cell differentiation and, as a result, tertiary dentin establishment in the exposed area while preserving tissue structure [6]. The effectiveness of a therapy is determined by the response of dental pulp cells to the material in direct contact with the tissue. Various pulp dressing medications have been implemented to maintain the health of the radicular pulp [7].

Mineral tri-oxide aggregate (MTA) is a bioactive and biocompatible self-setting hydrophilic calcium silicate cement which promotes the proliferation/differentiation of human dental pulp cells and showed calcified tissue-conductive activity, allowing for rapid dentine bridge construction and new hard tissue synthesis [8,9]. Dentin bonding systems have been investigated in humans and animals as viable direct capping materials in recent years as they have improved the capacity to bond to demineralized dentin tissues [10,11]. Tetric N-Bond Universal combines hydrophilic and hydrophobic material properties in one product. As a result, it achieves high bond strengths on wet as well as dry dentin surfaces. Even previous etching with phosphoric acid and overdrying of the surface does not compromise the bond to dentin. A Teteric N-Bond can also be used in dentin hypersensitivity and deep caries lesions, depending on the mode of application [12]. Self-etching has been used to create a self-assembled nanolayering of two 10-MDP (Methacryloyloxydecyl dihydrogen phosphate) molecules coupled by a stable MDP-Ca salt formation, which is more resistant to biodegradation. The less aggressive acidic treatment preserved the smear plugs into the dentinal tubules, thus decreasing the deteriorating effect of the pulpal pressure on the adhesive–dentin interface [13,14].

TheraCal PT (pulpotomy treatment) is a newly introduced biocompatible, dual-cured, resin-modified calcium silicate cement that acts as a barrier to the dental pulpal complex, preserving and promoting tooth viability. When TheraCal PT is directly applied to the pulp chamber, the setting of this dual-cured cement enables the immediate placement of restorative materials. After light curing the material for 10 s, clinicians can immediately place the desired adhesive with any bonding technique (self-, total-, selective-etch), base, and/or restoration [15]. Platelet-rich fibrin (PRF) is known to be enriched in platelets, leukocytes, and several growth factors, all of which play a significant role in cell proliferation and differentiation [16]. Studies have shown that the PRF membrane has a considerably slow sustained release of numerous essential growth factors for at least 1 week and up to 28 days, suggesting that the PRF membrane could release growth factors with its own biological scaffold for wound healing [17,18]. Dentin and pulp therapy may be possible due to dental progenitor cells found in pulp tissues [19].

Cytotoxicity screening assays are used to detect whether materials or their extracts promote cell death. A material’s effect on cell survival seems to be a determinant of biocompatibility. Cell culture determines in vitro cytotoxicity testing, which is a crucial component of bio-mechanical development [20]. A three-stage procedure has been recommended for evaluating the biocompatibility of dental materials. This involves the preliminary assessment of unspecific toxicity in cell lines, the growth inhibition of microorganisms, and the evaluation of genotoxicity, mutagenicity, and carcinogenicity in bacteria [21].

MTA, Tetric N-Bond Universal bonding agent, Theracal PT, and PRF have effects on human dental pulp tissues, either alone or in combination with other materials, and with differing success rates. These materials had not been compared for cell viability before this investigation. The purpose of this in vitro research study was to assess the functional differentiation potential of MTA, Tetric N-Bond Universal bonding agent, Theracal PT, and PRF placed in direct contact with human dental pulp stem cells (hDPSC) at 24 h, 48 h and 72 h. The null hypothesis was that MTA, Tetric N-Bond Universal bonding agent, Theracal PT, and PRF are cytotoxic to the human dental pulp stem cells (hDPSCs).

## 2. Materials and Methods

The study was conducted at the Institute of Dental Sciences in Bhubaneswar, Odisha, in collaboration with the Maratha Mandal Dental college and Research Centre in Belgaum, Karnataka. The IMS and SUM Hospital Siksha ‘O’ Anusandhan Ethics Committees approved the study and Institutional Ethical Clearance Ref. No.DRI/IMS.SH/SOA/2021/145.

### 2.1. Extraction and Culturing of hDPSC

The extraction of hDPSCs from the third molar was carried out according to “Goorha and Reiter’s” protocols [22]. Four healthy permanent third molars were extracted under aseptic conditions from healthy patients with no comorbid diseases. The extracted teeth were bored to gain access to the pulp chamber; using a barbed broach (Mani, INC, Tochigi, Japan) and a sharp spoon excavator (Mcare Instrument, XmsH, Maharashtra, India) the pulp tissue was extracted and hDPSCs were recovered within the first 48 h [22].

#### Culturing of hDPSC [7]

The obtained pulp stem was transported in Dulbecco’s Modified Eagle’s Medium (DMEM) (HiMedia Laboratories, Mumbai, Maharastra, India) for fibroblast culture. Isolation and harvesting of the hDPSC were performed as discussed in Bulbule et al. and Goorha and Reiter [7,22].

Using a sterile blade, pulp tissues were minced into small pieces in a petri dish and placed in a 37 °C washing medium (Delbecco’s phosphate buffered saline without calcium or magnesium and 1 percent antibiotic actinomycotic solution) to remove all debris. The pulp was placed in a centrifuge tube along with the washing media and spun at 800 rpm for 5 min at room temperature to produce a tissue pellet. The pellet was washed for 1 h at 37 °C in 4 mL of culture medium with 3 mg/mL of collagenase and 4 mg/mL of dispase II (Sigma-Aldreich, cat.no.4942078001), until the tissue underwent enzymatic dissociation. Cell pellets were obtained by enzymatically digesting the cell cultures in four steps and resuspending them in 5–10 mL of phosphate buffered saline. To create single-cell suspensions, these pellets were resuspended in 5 mL of DMEM and grown in 24-well microtiter plates for 7 days at 37 °C in 5% CO_2_ to obtain single-cell suspensions. Every three days, the culture media was replaced until cell confluence was attained. When the outgrown cells reached confluence, they were subcultured in growth media containing DMEM supplemented with 10% foetal bovine serum (FSB), 2 mmol L-1 of L-glutamine (a source of energy for rapidly dividing cells), and 1% penicillin, streptomycin, and amphotericin (PSA) (PAN-Biotech, Aidenbach, Germany) antibiotic solution to prevent bacterial contamination of the cell culture and incubated at 37 °C in a humidified atmosphere of 5% CO_2_ and 15% O_2_. Subculturing the outgrown cells at a 1:4 ratio generated hDPSCs. The cells were divided into five groups after the fourth passage based on test materials and a control group treatment.

### 2.2. Sample Preparation

In this investigation, hDPSCs were identified in the fourth linear passage and subsequently tested for viability. The negative control cells were the subcultured DMEM stem cells. Next, 5 mm × 2 mm diameter and height disc of test materials [MTA (MTA Plus, PrevestDenPro Limited, Jammu, India), Tetric N-Bond Universal bonding agent (Ivoclar vivadent, Schaan, Liechtenstein), Theracal PT (Bisco Inc., Schaumburg, IL, USA)] are prepared using sterilized cylindrical rubber moulds (UV radiation, 15 min) and stored in an incubator (Sigma Instruments, Mumbai, Maharashtra, India) (37 °C, 48 h) to ensure complete setting. As per the manufacturer’s instructions, one scoop of MTA powder and a drop of distilled water were mixed for 30 s to obtain a homogeneous consistency. The mixed MTA is poured into the rubber moulds and allowed to set completely. Tetric N-Bond Universal (drops placed in the rubber moulds) and premixed Theracal PT were injected into different rubber moulds and photocured (1200 mW/cm^2^ for 20 s) with a light-emitting diode curing lamp (Bluephase 20i; Ivoclar Vivadent, Schaan, Liechtenstein). The sample discs were immersed in respective Dulbecco’s Modified Eagle’s Medium and grouped as Group I-MTA, Group II-Tetric N-Bond Universal bonding agent, and Group III-Theracal PT. All three groups were kept for 24 h at 37 °C and 5% CO_2_ in a humid atmosphere.

Preparation of PRF (Group IV): 10 mL of healthy participants’ venous blood was collected and split into 5 mL sterile vacuum blood tubes without anticoagulants. Blood samples were centrifuged at 3000 rpm for 15 min. PRF was made before introducing test materials into cultured pulp stem cells. The control group (Group V) is a subcultured (DMEM) hDPSC, and no test materials were added.

### 2.3. Cytotoxicity Assay

MTA, Tetric N-Bond Universal, and Theracal PT, each weighing one thousand milligrams, were dissolved in one thousand microliters of dimethyl sulfoxide to produce 100 percent solutions. Using Eppendorf tubes, 100 mL of the test agents and the control were administered in triplicate to 96-well microtiter plates containing 2 × 10^5^ cells/well of the stem cell culture. For the PRF group, the PRF was cut into 1 × 1 mm pieces using a scalpel and applied directly in triplicate. The number of viable cells after 24-, 48-, and 72-h incubation at 37 °C in a humidified environment of 5% CO_2_ and 95% air was evaluated using the MTT (3-[4,5-dimethylthiazol-2-yl]-2,5 diphenyl tetrazolium bromide) assay [7].

#### MTT Assay

MTT solution preparation (stock solution): At pH 7.4, 5 mg of MTT powder was dissolved in 1 mL of Dulbeco’s phosphate buffered saline (PBS). MTT solution was filtered via a 0.2 micron filter and stored in a light-resistant, sterile container. Minimum Essential Media (MEM) with 10% heat-inactivated foetal calf serum (FCS) and 5% of a mixture of Gentamicin (10 μg), Penicillin (100 units/mL), and Streptomycin (100 g/mL) in the presence of 5% of CO_2_ were used to maintain the stem cell lines (hDPSC) in 96-well microtiter plates at 37 °C for 48–72 h.

In this investigation, 10, 20, 25, 30, and 50 µL of the stock solution (10 mg/mL produced in dimethyl sulfoxide (DMSO)) were applied to wells containing 100 µL of medium, and the final concentrations were 10, 20, 25, 30, and 50 µg/mL. After 48 h of incubation at 37 °C in a humidified atmosphere containing 5% CO_2_, a stock solution of MTT was added to each well (20 µL, 5 mg per ml in sterile PBS) for an additional 4 h. The supernatant was aspirated carefully, and 100 µL of DMSO was added to dissolve the “Formazan blue” crystals that had precipitated, and the optical density (OD) at 570 nm was measured using a spectrophotometer. The values are the mean of five readings. The concentration at which treated cells’ OD decreased by 50% compared to the untreated control. For accuracy, the OD was measured in triplicate at 540 and 720 nm, and the mean was used. The live cell percentage is calculated as follows: Surviving cells (%) = Mean OD of test compound × 100/mean OD at control [23].

To evaluate cell viability in all groups after 24 h, 48 h, and 72 h, 20 μL of 5 mg/mL MTT was added to fifteen wells (three wells of 100 μL of 100% MTA, three wells of 100 μL of 100% Tetric N-Bond Universal, three wells of 100 μL of 100% Theracal PT, three wells of 100 μL of 100% PRF, and three wells of cells without treatment) on a 96-well microtiter plate and incubated at 37 °C, 5% CO_2_, with 98% humidity for 4 h. At the end of the incubation time, the MTT-containing media was withdrawn and 100 μL of dimethyl sulfoxide was added to each well. The purple MTT formazan was dissolved utilizing a microplate shaker. Dental pulp stem cell viability was expressed as the colour intensity of experimental wells relative to control. The same method was used three times to confirm the readings. On a microplate reader, absorbance was measured at 492 nm with background subtraction at 620 nm (Lisaplus, Aspen Diagnostics Pvt. Ltd., Mumbai, Maharashtra, India). Evaluations at 48 and 72 h were conducted using the same method.

## 3. Results

Descriptive and analytical statistics were computed in SPSS Version 24.0 (IBM Corporation, Chicago, IL, USA). The Shapiro–Wilk test verified the normality of the data. The one-way ANOVA test compared group mean differences, and post-hoc analysis used Tukey’s HSD (Honest Significant Difference). A paired sample *t*-test (*p* 0.05) was used to assess intra-group mean differences. The cytotoxicity was measured in percentage as shown in Table 1.

### 3.1. Comparison of MTT Assay Results at 24 h

One-way ANOVA showed statistically significant (*p* < 0.001) differences in mean OD values among the groups. The group IV had the highest mean OD values of 0.575 ± 0.003, followed by group III (0.526 ± 0.002), group I (0.5464 ± 0.003), group V (0.435 ± 0.004), and group II (0.391 ± 0.007) (Figure 1). The rate of cell proliferation at 24, 48, and 72 h (Tukey’s HSD test) showed statistically significant differences (*p* < 0.001) between pairwise comparisons shown in Table 2.

#### 3.1.1. At 48 h

The group IV had the highest mean OD values of 0.627 ± 0.006, followed by group I (0.576 ± 0.004), group III (0.527 ± 0.001), group V (0.455 ± 0.003), and group II (0.423 ± 0.004) (Figure 2 and Table 1 and Table 2).

#### 3.1.2. At 72 h

The group IV had highest mean OD values of 0.575 ± 0.003, followed by group III (0.564 ± 0.002), group I (0.456 ± 0.004), group V (0.399 ± 0.001), and group II (0.382 ± 0.002) (Figure 3 and Table 1).

### 3.2. Intra Group Comparison of MTT Assay Results

Groups were compared at 24 versus 48 h, 48 versus 72 h, and 24 versus 72 h. Group I was significantly higher (*p* = 0.00 1) at 48 h (0.575 ± 0.004) than 24 h (0.464 ± 0.003), and the 72 h results (0.456 ± 0.004) were significantly lower (*p* 0.001) than the 48 h (0.575 0.004). In group II: results at 48 h (0.423 ± 0.004) were significantly higher (*p* = 0.038) than at 24 h (0.391 ± 0.007) and, at 72 h (0.382 ± 0.002), they were significantly lower (*p* = 0.001) than 48 h. Group III: There was an insignificant difference (*p* = 0.580) at 24 and 48 h; however, 24 h (0.526 ± 0.002) was significantly higher (*p* = 0.001) than 72 h (0.564 ± 0.002). Group IV: 48 h (0.627 ± 0.006) was significantly higher (*p* = 0.012) than at 24 (0.575 ± 0.003) and 72 h (0.575 ± 0.003). In group V: at 72 h (0.399 ± 0.001) there was a significant decrease than at 24 (0.435 ± 0.004) and 48 h (0.455 ± 0.003), as shown in Figure 4.

Expect for group II, the viable cells were more at 48 h for all the groups. The cell death was highest at 24 h and decreased significantly at 48 h and 72 h (Table 1).

## 4. Discussion

Conventional pulp capping agents (PCA)/regenerative endodontic therapies recommend treating pulp–dentine wounds with calcium hydroxide to preserve damaged tissue, induce reparative dentine production, and sustain pulp. Advances in material science have led to improved pulp capping materials for deciduous and permanent teeth [24]. DPC heals reversibly damaged pulp by stimulating dentin bridge formation and restoring pulp-dentin structure and function. A bioactive substance (PCA) stimulates undifferentiated mesenchymal stem cells to become odontoblastoid, osteoblastoid, or cementoblastoid cells, which repair wounds [25]. In vitro cytotoxicity and functional differentiation experiments with human dental pulp cells are effective for analyzing cell reactivity, behavior, and fate when exposed to test materials, supporting animal-free research methods [26,27]. In this study, hDPSCs were chosen because they are directly generated from neural crest cells, are less mature, proliferate faster, and form dentin-forming odontoblasts. This cell can self-renew by splitting into multiple lineages. Because third molar roots are often absent by the age of 18, these teeth have a large pool of undifferentiated cells in the dental germ pulp (cell rich zone) [28,29]. Odontoblasts, apical vasculature, and periodontal ligament are mesenchymal components of teeth [30]. In this study, stem cells suspended in DMEM with 10% foetal bovine serum (FSB) and 2 mmol L^−1^ L-glutamine were used as a negative control. Instead of directly culturing cells with test materials, they were suspended in test-containing fluid to reduce the cytotoxic effects of test materials on stem cells. This experimental setting resembles clinical conditions in which dental pulp cells are near blood vessels and nerve fibers that receive solubilized capping materials. Only viable hDPSCs grown in standard medium were collected, showing functional differentiation.

MTT was used to examine the effects of bioinductive compounds on hDPSC viability and proliferation. Mitochondrial succinate dehydrogenase decreased the amount of yellow 3-(4,5-dimethythiazol-2-yl)-2,5-diphenyl tetrazolium bromide. MTT transforms into an insoluble, dark purple formazan in the mitochondria [8,31]. During the method, viable cells were solubilized in an organic solvent (DMSO, Isoproanol, 3-(4,5-dimethylthiazol-2-yl)-2,5-diphenyltetrazolium bromide) and released formazan. Solubilized formazan was measured with a spectrophotometer [32].

The MTT assay measures the metabolic activity of stem cells in culture conditions over a set period of time. As the absorbance value rises over time, it might be read as a higher level of cellular activity and viability, and thus as an indirect measure of cell proliferation. Intra-group comparisons of MTT assays were performed for individual groups at 24 h, 48 h, and 72 h to obtain the best results. The results suggested that, in all time periods, the cell proliferation rate was the significant. Moreover, the 72-h MTT assay results for Theracal PT were substantially higher than the 24- and 48 h results. This suggested that cell growth was constant across time. After 24 and 72 h, Group IV (PRF) had the highest mean OD value, followed by group III (Theracal PT), group I (MTA), group V, and group II (Tertric N-Bond). In a recent work by Rodrguez-Lozano et al. (2021) [33], MTA demonstrated higher cell viability than Theracal PT in an MTT experiment. MTA may cause a thicker hard tissue bridge, less inflammation, and pulp tissue necrosis (less caustic impact) quickly after application [8]. A study compared the cell viability and cytotoxicity of bioinductive materials (MTA, Biodentine, and Theracal LC) for essential pulp capping and concluded that Theracal PT has the ability to form dentin bridges [34,35]. In the set material, alkalinity is removed and cytotoxicity is reduced; therefore, the cellular response to MTA is influenced by a variety of factors, including the cell type and the study duration [8]. This could be one of the reasons why MTA performed better at 24, 48, and 72 h. Studies have suggested that MTA increases the formation of cytokines in human osteoblasts, allowing the cells to bond well to substances and therefore play a part in the creation of dentin bridges [36,37].

In the current study, group II (Tetric N-Bond Universal bonding agent) showed the least viability of cells (Figure 4). Cell death was 10% at 24 h on direct contact with hDPSC, and subsequently decreased at 48 and 72 h. Tetric N-Bond Universal contains methacrylates, ethanol, water, and highly dispersed silicon dioxide might attribute for cell death. The thickness or amount of bonding agent may be a factor in the increase in cell death. As per previous studies, it contains HEMA (2-hydrxyethyl methacrylate), D3MA (decanediol dimethacrylate; hydrophobic) and MCAP (methacrlated carboxylic acid polymer) [12], which are released in the first 24 h after polymerization, and gradual degradation starts at later stages [38]. Our results showed a gradual decrease in the cytotoxicity with progression of time.

Generally, dentin bonding hybridization and adhesive diffusion into dentin tubules may protect the dentin from bacterial leakage, reducing recurrent pulpal inflammation [11,14]. A self-assembled nanolayering of two 10-MDP molecules bonded by a stable MDP-Ca salt formation creates an adhesive interface that is more resistant to biodegradation [39]. The interaction of a polyalkenoic acid copolymer with the calcium in hydroxyapatite is another chemical process [14]. A study has shown that dentin bonding agent (DBA) can cause an inflammatory infiltration in the tissues around the exposure site; moreover, close to the pre-dentin, foreign body type giant cells were evident [14]. No signs of cellular differentiation and dentin neoformation in the proximity of the injured area were observed at day 30 after exposure [40]. Camphoroquinone (CQ) is a widely used aliphatic kind of photosensitizer that has been linked to free radicals, particularly reactive oxygen production. Camphoroquinone (CQ) leached from DBA may be a reason for its cytotoxic effect as it is a mutagen and cytotoxic agent [10]. Tetric N-Bond Universal does not contain CQ; instead, it has a patented molecule. Alexandre M. Fernandes et al. compared direct pulp capping of human pulp with calcium hydroxide (CH) and DBA and found no expression of type III collagen and fibronectin in the SBAS group [35,41]. A study evaluated clinically and histologically human dental pulp tissue response after direct pulp capping using an adhesive bonding agent and calcium hydroxide on twenty-eight caries free 3rd molar teeth. They observed histological signs of pulpitis that non-significantly increased and a weaker thin mineralized tissue layer [41]. At 72 h, cytotoxicity was found to be insignificant in group II. However, in a clinical situation, the amount of boning agent and area application varied considerably from group to group. Due to the presence of odontoblastic processes and undifferentiated mesenchymal cells in dentin, the newer universal system is anticipated to be non-cytotoxic and to maintain cell viability.

Group IV showed maximum viable cells, as platelet concentrates (PRF) contain growth factors that promote and modify cellular functions, as a pulp capping agent plays a significant role in hard and soft tissue healing, exhibiting chemotactic and mitogenic properties that stimulate and change cellular processes [42]. They have the ability to transform dental pulp cells into odontoblastic-like cells, resulting in the production of reparative dentin. PRF has been shown to have effects associated with the inhibition of inflammation in hDPSCs exposed to lipopolysaccharide (LPS). Growth factors (BMP(Bone morphogenetic proteins), FGF(Fibroblast growth factor), and TGF(Transforming growth factor)) control dentin, cementum, and bone growth in stem cells from the dental pulp and periodontal ligament [43]. Reparative dentinogenesis induces and enhances stem cell differentiation into odontoblast-like restructure for both growth factors and bioinductive materials. Platelet-rich fibrin (PRF) belongs to the second generation of platelet concentrate products, named Choukroun’s PRF after its inventor. PRF can be considered as an immune concentrate with a specific composition and a 3D architecture. It consists of an intimate assembly of cytokines, glycan chains, and structural glycoproteins enmeshed within a slowly polymerized fibrine network [42,44].

The biocompatibility of PRF on hDPCs has been confirmed in the present in vitro study. PRF was as effective as the widely accepted MTA when in direct contact with the cells, and were even superior to MTA in promoting cell proliferation [45]. Chen et al. studied the cytobiology of PRF on dental pulp stem cells. PRF stimulated the differentiation of hDPSC to odonto-/osteoblastic fates by increasing the expression of the alkaline phosphatase (Alp), dentin sialophosphoprotein (Dspp), dentin matrix protein 1(Dmp1), and bone sialoprotein (Bsp) genes [46]. A previous in vivo study demonstrated that PRF can be used as a scaffold to support the dental stem cells’ transplantation and induce the de novo formation of dentin-like and pulp-like tissue. This study also suggests that the presence of bioactive materials does not hinder or impede the formation of new hard tissues; however, different biomaterials may promote greater mineralized tissue deposition [2]. This study justified cell differentiation to determine whether these materials are safe for pulp capping and regenerative therapy.

The study’s limitation is that the cell viability and cytotoxicity of pulp capping agents (PCA) were investigated in an in vitro controlled environment with no inflammation in the cells being treated. Further clinical studies can be conducted to determine the efficacy of inflamed pulp caps.

## 5. Conclusions

To summarize, PRF demonstrated greater cell viability than MTA at all intervals. Theracal PT bonding agent preserved the hDPSC better than the Tetric N-Bond Universal bonding agent. Cytotoxicity (cell death) was only observed in the Tetric N-Bond Universal bonding agent group. More research is needed to determine the effect of the dentin bonding agent on the viability of pulp tissue cells.

## Figures and Tables

**Figure 1 jcm-12-00539-f001:**
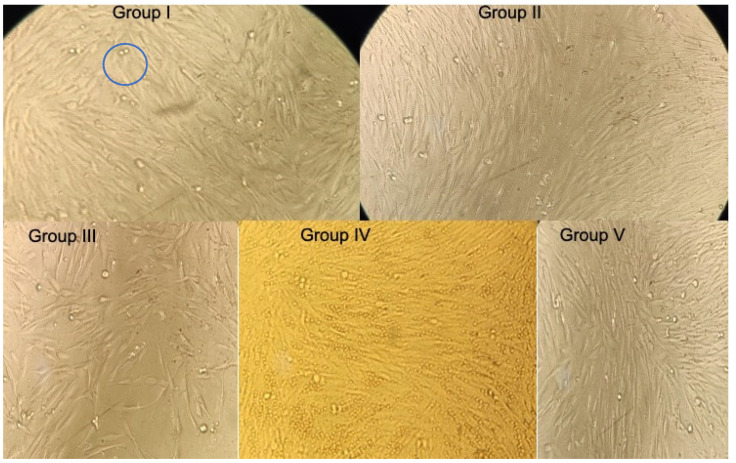
Histological assessment of MTT (3-[4,5-dimethylthiazol-2-yl]-2,5 diphenyl tetrazolium bromide) Assay of hDPSCs (Human Dental Pulp Stem Cells) treated with groups after 24 h. Circle represents living stem cells in cluster form.

**Figure 2 jcm-12-00539-f002:**
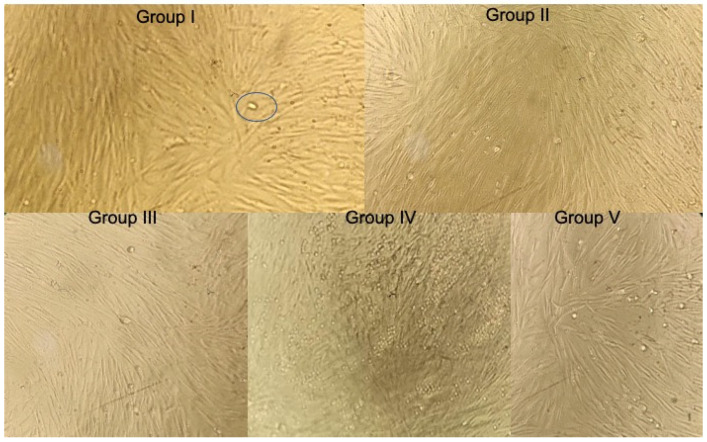
Histological assessment of MTT Assay of hDPSCs treated with groups after 48 h. Circle represents living stem cells in cluster form.

**Figure 3 jcm-12-00539-f003:**
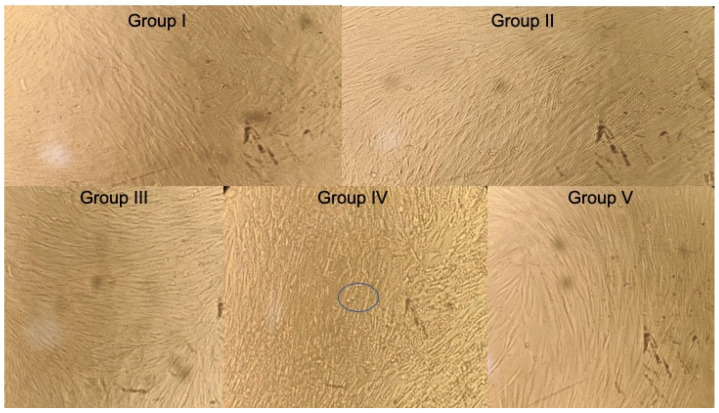
Histological assessment **of** MTT assay of hDPSCs treated with groups after 72 h. Circle represents living stem cells in cluster form.

**Figure 4 jcm-12-00539-f004:**
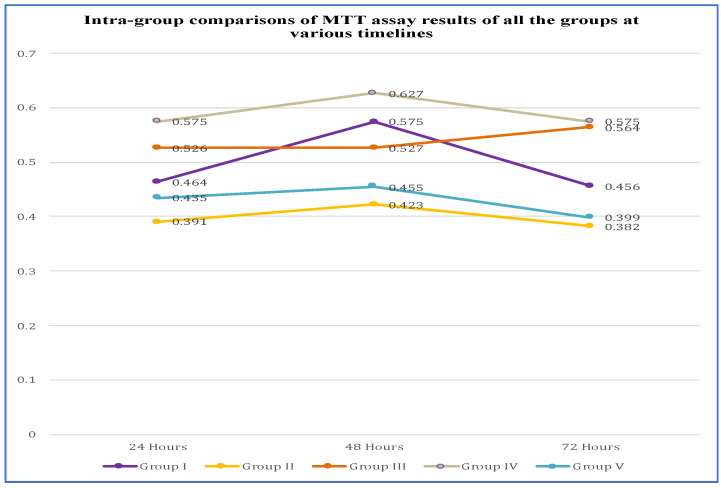
Intra group comparison of MTT assay results.

**Table 1 jcm-12-00539-t001:** MTT assay results (cell viability and cytotoxicity) for groups at 24, 48, and 72 h.

	24 h				48 h				72 h			
Groups	OD Values	Mean OD	Cell Proliferation (%)	Cell Death (%)	OD Values	Mean OD	Cell Proliferation (%)	Cell Death (%)	OD Values	Mean OD	Cell Proliferation (%)	Cell Death (%)
Group I	0.464	0.464	106%	00	0.577	0.576	126%	00	0.459	0.456	114%	00
	0.462				0.579				0.458			
	0.468				0.571				0.451			
Group II	0.398	0.391	90%	10%	0.420	0.423	92%	8%	0.382	0.382	95%	5%
	0.392				0.422				0.381			
	0.384				0.428				0.385			
Group III	0.525	0.526	120%	00	0.529	0.527	115%	00	0.561	0.564	141%	00
	0.529				0.528				0.565			
	0.526				0.526				0.566			
Group IV	0.579	0.575	132%	00	0.620	0.627	137%	00	0.575	0.575	144%	00
	0.575				0.629				0.579			
	0.572				0.632				0.572			
Group V	0.431	0.435	100%	00	0.452	0.455	100%	00	0.399	0.399	100%	00
	0.436				0.459				0.401			
	0.439				0.454				0.398			

OD: optical Dentsitry.

**Table 2 jcm-12-00539-t002:** Comparison of MTT assay results at 24, 48, and 72 h among the groups.

	24 h					48 h				72 h			
Groups	N	Mean	S.D.	F-Value	*p*-Value	Mean	S.D.	F-Value	*p*-Value	Mean	S.D.	F-Value	*p*-Value
Group I	3	0.464	0.003	873.382	<0.001 ^†^	0.576	0.004	1186.55	<0.001 ^†^	0.456	0.004	2716.54	<0.001 ^†^
Group II	3	0.391	0.007			0.423	0.004			0.382	0.002		
Group III	3	0.526	0.002			0.527	0.001			0.564	0.002		
Group IV	3	0.575	0.003			0.627	0.006			0.575	0.003		
Group V	3	0.435	0.004			0.455	0.003			0.399	0.001		

*p*-value derived from one-way ANOVA test: ^†^ significant at *p* < 0.05.

## Data Availability

Requested data from corresponding author.

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
