# Peer review of "Analysis of Pulp Tissue Viability and Cytotoxicity of Pulp Capping Agents"

_jcm, 2023, doi:10.3390/jcm12020539_

Round 1

Reviewer 1 Report

The results of this study mostly approve facts that are already known and shown in the literature. The efficiacy of tested materials in these analyses about pulp tissue viability is high. Theracal PT and MTA (calcium silicate based materials) and PRF (contains numerous growth factors) have bioinductive potential of forming dentine bridge, with absence of inflammation, or weak inflammation after direct pulp capping.

Methodology and results are clearly presented and discussion  clearly discussed with recent references.

Literature is up to date.

Author Response

Dear Reviewer,

we appreciate the effort in reviewing our manuscript and your comments and suggestions will substantial improves the quality of manuscript.

"The results of this study mostly approve facts that are already known and shown in the literature. The efficiacy of tested materials in these analyses about pulp tissue viability is high. Theracal PT and MTA (calcium silicate based materials) and PRF (contains numerous growth factors) have bioinductive potential of forming dentine bridge, with absence of inflammation, or weak inflammation after direct pulp capping"

Although there is less evidence in the literature for theraCal PT and Tetric N bonds, cytotoxicity on hDPSC with different intervals has been found to be significant. The study suggests comparing between groups at various intervals.

Reviewer 2 Report

Dear authors,

Your article Analysis of Pulp Tissue Viability and Cytotoxicity of Pulp Cap- 2 ping Agents” is of insufficient quality.  In the M&M section the use of “pulpal stem cells” was described. Unfortunately, the authors did not explain how these stem cells were extracted, identified and cultured. A statistical section is missing, too. The use of control samples is not clear. Only the MTT assay was used to determine cyto-toxicity. In the figure standard division is missing as well as significance was not marked. Discussion does not explain how own results need to be interpreted in the context of literature currently available. Further, either use DPSC or HDPS or hDPS in your study.  Please carefully check if all abbreviations are defined in prior! Consider to rework your study in accordance to the remarks given.

Author Response

Dear Reviewer,

We greatly appreciate your time and effort in reviewing our manuscript and making suggestions to improve its quality.

Ponit 1: n the M&M section the use of “pulpal stem cells” was described. Unfortunately, the authors did not explain how these stem cells were extracted, identified and cultured.

We have discussed the procedure of extraction and culturing procedure. Marked it red in the manuscript.

Point 2: A statistical section is missing, too. The use of control samples is not clear.

highlighted the statistical section and uniformity of control group as group V is mentioned in entire manuscript.

Point 3: Only the MTT assay was used to determine cyto-toxicity.

yes 

Point 4: In the figure standard division is missing as well as significance was not marked.

Highlighted in table 2

Point 5: Discussion does not explain how own results need to be interpreted in the context of literature currently available.

modified and interpreted 

Point 6: Further, either use DPSC or HDPS or hDPS in your study.  Please carefully check if all abbreviations are defined in prior

Uniformity is maintained and abbreviations are defined in prior.

Reviewer 3 Report

This study assessed, in vitro, the cytocompatibility of four materials with potential clinical application in Vital Pulp Therapy. The study is under the scope of this journal and the methodology used is adequate. However, the manuscript needs substantial improvement to be scientifically sound.

My main concern is with the fact that authors are comparing 4 groups with very different properties (2 calcium-silicate cements, one adhesive system and a biological scaffold). The introduction is lacking the rationale for comparison of those 4 materials, and it is crucial to present that reasoning. Moreover, clinical studies at the last decade of the 20th century demonstrate really bad results with the use of adhesive systems for direct pulp capping. Therefore, inclusion of this group needs justification, and also the amount of the adhesive resin used in this study does not correlate with the amount used in a clinical scenario, contrasting with the other 3 tested materials, which needs to be addressed at discussion.

Some specific concerns:

English writing needs correction by proficient professionals;

Introduction

The sentence “Vital Pulp Therapy (VPT) are “protective liners, indirect pulp therapy, direct pulp therapy, pulpotomy, and apexogenesis.” needs correction of the content - VPT is a term applied to a group of treatment approaches which includes, direct pulp capping, atrial and full pulpotomy, and apexogenesis (materials should not be mixed-up with treatment modalities);

 Reference number 2 is not adequate to support the statement “The most important characteristics of these materials are to promote tissue healing, cytocompatibility, and the ability to seal the lesion” because sealing is not addressed in an in vitro study. An in vivo study must be quoted to support this sentence;

First paragraph of introduction is quoting only in vitro studies to support characteristics that can only be observed in vivo, therefore, better references need to be cited;

Second paragraph of introduction is too generic without any relevant information for this manuscript, is just general concepts from a textbook.

Materials and methods

Description of the groups and correspondence with the group numbers needs to be clarified.

Results

Groups nomenclature need to be consistent, in one part we have 4 groups and a control, other parts of the manuscript five groups… 

Intragroup comparisons are useless in the way they are presented. It is a lot of information without relevance and distracting the readers. Please describe just the results that are relevant for interpretation.

In Figure 4 correspondence between group numbers and materials tested needs to be presented at the legend, to be self-explanatory.

Discussion

P8L267 “high success rate??”

P9L317-326, a previous in vivo study demonstrated that PRF can be used as a scaffold to support the Dental Stem Cells transplantation and induce de novo formation of dentin-like and pulp-like tissue. This study also suggests that the presence of bioactive materials does not hinder or impede the formation of new hard tissues, but different biomaterials may promote greater mineralized tissue deposition (Sequeira et al 2021 Regeneration of pulp-dentin complex using human stem cells of the apical papilla: in vivo interaction with two bioactive materials. Clinical Oral Investigations). Information of the present study needs to be interpreted in the light of the previously indicated study data.

Conclusion

This is an in vitro study and conclusions must be limited to in vitro research, no clinical indications can be drawn from the results of this study.

Author Response

Dear Reviewer

We sincerely appreciate your time and effort in reviewing our manuscript, and your suggestions significantly improve it.

Point 1: My main concern is with the fact that authors are comparing 4 groups with very different properties (2 calcium-silicate cements, one adhesive system and a biological scaffold). The introduction is lacking the rationale for comparison of those 4 materials, and it is crucial to present that reasoning.

Inclusion and modification is done as per the suggestion. Highlighted in the manuscript

Point 2: Moreover, clinical studies at the last decade of the 20th century demonstrate really bad results with the use of adhesive systems for direct pulp capping. Therefore, inclusion of this group needs justification, and also the amount of the adhesive resin used in this study does not correlate with the amount used in a clinical scenario, contrasting with the other 3 tested materials, which needs to be addressed at discussion.

justified regarding the concern raised in discussion.

Point3: The sentence “Vital Pulp Therapy (VPT) are “protective liners, indirect pulp therapy, direct pulp therapy, pulpotomy, and apexogenesis.” needs correction of the content - VPT is a term applied to a group of treatment approaches which includes, direct pulp capping, atrial and full pulpotomy, and apexogenesis (materials should not be mixed-up with treatment modalities);

The sentence is removed. discussed as per the study protocol.

Point 4: Reference number 2 is not adequate to support the statement “The most important characteristics of these materials are to promote tissue healing, cytocompatibility, and the ability to seal the lesion” because sealing is not addressed in an in vitro study. An in vivo study must be quoted to support this sentence;

Inclusion of reference are made respectively.

Point 5:First paragraph of introduction is quoting only in vitro studies to support characteristics that can only be observed in vivo, therefore, better references need to be cited;

Second paragraph of introduction is too generic without any relevant information for this manuscript, is just general concepts from a textbook.

modified as per suggestion and highlighted.

Point 6: Description of the groups and correspondence with the group numbers needs to be clarified.

Clarified and maintained consistently throughout the manuscript.

Point 7: Groups nomenclature need to be consistent, in one part we have 4 groups and a control, other parts of the manuscript five groups… 

Intragroup comparisons are useless in the way they are presented. It is a lot of information without relevance and distracting the readers. Please describe just the results that are relevant for interpretation.

In Figure 4 correspondence between group numbers and materials tested needs to be presented at the legend, to be self-explanatory.

group consistency is maintained, changed the figure 4 and simplified the intergroup interpretation.

Point 8: P8L267 “high success rate??”

P9L317-326, a previous in vivo study demonstrated that PRF can be used as a scaffold to support the Dental Stem Cells transplantation and induce de novo formation of dentin-like and pulp-like tissue. This study also suggests that the presence of bioactive materials does not hinder or impede the formation of new hard tissues, but different biomaterials may promote greater mineralized tissue deposition (Sequeira et al 2021 Regeneration of pulp-dentin complex using human stem cells of the apical papilla: in vivo interaction with two bioactive materials. Clinical Oral Investigations). Information of the present study needs to be interpreted in the light of the previously indicated study data

Included as per the suggestion

This is an in vitro study and conclusions must be limited to in vitro research, no clinical indications can be drawn from the results of this study

Modified as per the suggestion and highlighted.

Round 2

Reviewer 3 Report

Authors improved the quality of the manuscript.